# Posterior Urethral Valves, Unilateral Vesicoureteral Reflux, and Renal Dysplasia (VURD) Syndrome: Long-Term Longitudinal Evaluation of the Kidney Function

**DOI:** 10.3390/ijerph20136238

**Published:** 2023-06-27

**Authors:** Irene Paraboschi, Adele Giannettoni, Guglielmo Mantica, Alexios Polymeropoulos, Pankaj Mishra, Joanna Clothier, Massimo Garriboli

**Affiliations:** 1Pediatric Nephro-Urology, Evelina London Children’s Hospital, Guy’s and St Thomas’ NHS Foundation Trust, London SE1 7EH, UK; 2Department of Pediatric Urology, Fondazione IRCCS Cà Granda Ospedale Maggiore Policlinico, Via Francesco Sforza 28, 20122 Milan, Italy; 3Department of Urology, IRCCS Ospedale Policlinico San Martino, 16132 Genoa, Italy; 4Department of Statistics ad Quantitative Methods, University of Milano-Bicocca, 20126 Milan, Italy; 5Stem Cells & Regenerative Medicine Section, Developmental Biology & Cancer Programme, UCL Institute of Child Health, London WC1N 1EH, UK

**Keywords:** posterior urethral valves, VURD syndrome, pop-off mechanism, vesicoureteral reflux, children

## Abstract

The presence of unilateral vesicoureteral reflux (VUR), and renal dysplasia associated with posterior urethral valves (PUV) (VURD syndrome) was believed to represent a pressure-released pop-off mechanism protecting kidney function. We aimed to investigate its role with respect to long-term kidney function in a cross-sectional and longitudinal analysis. We compared the iohexol glomerular filtration rate (GFR) measured at 5 (GFR_5_) and 10 (GFR_10_) years of age in children with (Group A) and without (Group B) VURD syndrome, who underwent PUV resection under 2 years of age. VURD syndrome was diagnosed in cases of unilateral loss of kidney function (<15% on nuclear medicine test) associated with ipsilateral grade IV-V VUR. VURD syndrome was diagnosed in 16 (12.8%) out of 125 patients who met the inclusion criteria. While the median GFR_5_ was similar in the 2 groups [Group A: 87.3 (74.7–101.2) mL/min/1.73 m^2^ vs. Group B: 99.6 (77–113) mL/min/1.73 m^2^, *p*-value: 0.181], the median GFR_10_ values were significantly lower in children with VURD syndrome [Group A: 75.7 (71.2–85.9) mL/min/1.73 m^2^ vs. Group B: 95.1 (81.2–114.2) mL/min/1.73 m^2^, *p*-value: 0.009]. Similar results were obtained in a longitudinal analysis of the children with GFR measurement available both at 5 and 10 years of age [GFR_5_ in Group A: 93.1 (76.9–103.5) mL/min/1.73 m^2^ vs. Group B: 97.5 (80–113) mL/min/1.73 m^2^, *p*-value: 0.460; GFR_10_: Group A: 71.9 (71.9–85.9) mL/min/1.73 m^2^ vs. Group B: 94.8 (81.5–110.6) mL/min/1.73 m^2^, *p*-value: 0.024]. In conclusion, VURD syndrome does not show a protective role in kidney function preservation. On the contrary, it seems to be associated with a deterioration of the kidney function on a long-term follow-up.

## 1. Introduction

Posterior urethral valves (PUV) are congenitally obstructing membranous folds of the male posterior urethra [1]. They represent the most common cause of bladder outlet obstruction in boys, affecting approximately 1 in 3800 male births per annum [2].

Despite recent progress in prenatal assessment and postnatal management, PUV still represent one of the greatest challenges for pediatric urologists and nephrologists, with a quarter of children progressing to end-stage kidney failure, mainly due to the high back pressure transmitted to the developing kidneys [3].

Three main anatomical variables have been described to act as a pressure-release pop-off mechanism and supposedly contribute to the preservation of the overall kidney function [4,5,6]. These include large congenital bladder diverticula, urinary extravasation and/or urinoma presenting in the newborn period and the association of PUV, unilateral high-grade vesicoureteral reflux (VUR), and ipsilateral renal dysplasia (referred to as VURD syndrome) [4].

The potential effect of VURD is to protect the function of a single renal unit by dissipating the pressure created in the obstructed bladder toward the contralateral kidney and ureter, together this renal unit would act as an extra-reservoir, but at the expense of kidney function in that kidney.

Since its first description by Hoover and Duckett in 1982 [7], several studies have investigated the effects of VURD syndrome on long-term kidney outcomes in children with PUV, however, with contradictory results.

We aim to explore the role of VURD syndrome in protecting overall kidney function in our cohort of PUV children focusing on long-term cross-sectional and longitudinal iohexol GFR results.

## 2. Materials and Methods

We retrospectively reviewed the medical records of boys diagnosed with PUV between January 2001 and January 2022 in a single tertiary care pediatric center.

Exclusion criteria were PUV ablation performed after 2 years of age (we considered late-diagnosed patients representing a different population) and missing data (preoperative micturating cystourethrogram (MCUG) and dimercaptosuccinic acid (DMSA) or mercaptoacetyltriglycine (MAG3) study performed within the first few months of life and distant from episodes of acute urinary tract infection).

Children were divided into 2 groups for comparison based on the presence of pre-ablation VURD syndrome (Group A patients with VURD syndrome and Group B without VURD syndrome).

VURD syndrome was defined as the presence of unilateral high-grade VUR (grade IV–V) demonstrated on MCUG associated with ipsilateral dysplastic and poorly functioning kidney (<15%) identified on nuclear medicine scan performed at diagnosis.

Values of Glomerular filtration rate (GFR), measured by iohexol plasma clearance at 5 (GFR_5_) and 10 (GFR_10_) years of age, routinely performed as part of our protocol, were collected and used for the assessment of the long-term kidney function.

Initial analysis included all patients meeting the inclusion criteria (cross-sectional analysis) and subsequently, we performed a longitudinal evaluation, only including patients with available data at both time points GFR (5 and 10 years of age) (longitudinal analysis).

Several additional variables were recorded in the 2 groups of patients, including the rate of antenatal diagnosis, the age of valve ablation, and the nadir serum creatinine levels.

Descriptive statistics were performed for all the variables. For quantitative variables summary statistics were reported as median and 1st–3rd interquartile (q_1_–q_3_) whereas for qualitative variables summary statistics were based on absolute and relative frequencies. Comparisons between the 2 groups of patients were performed using the non-parametric Mann-Whitney U-test or the Wilcoxon signed ranks test on a set of matched samples. Frequency data were analyzed using the Pearson chi-square test, or Fisher exact test in cases of expected frequencies lower than 5. Statistics were performed using Minitab statistical software (version 19) by a professional statistician (A.P).

## 3. Results

Among a total of 336 boys diagnosed with PUV in the study period and followed up in our tertiary care pediatric center, 125 (37.2%) children met the inclusion criteria. Of these, 16 (12.8%) children were diagnosed with VURD syndrome (Figure 1).

As shown in Table 1, the 2 populations of children with (Group A, *n* = 16) and without (Group B, *n* = 109) VURD syndrome were comparable with regard to the rate of antenatal diagnosis [Group A: *n* = 8 (50.0%) vs. Group B: *n* = 65 (59.6%), *p*-value: 0.589], age at valve ablation [Group A: 48 (30–230.5) days vs. Group B: 33 (16–108) days, *p*-value: 0.093], and nadir serum creatinine levels [Group A: 26.5 (22.5–34.0) µmol/L vs. 26.0 (22.0–33.0) µmol/L, *p*-value: 0.935].

As shown in Figure 2, on univariate analysis, no statistically significant difference was identified between the 2 groups of patients about GFR values measured at 5 years of age [Group A: 87.3 (74.7–101.2) mL/min/1.73 m^2^ vs. Group B: 99.6 (77–113) mL/min/1.73 m^2^, *p*-value: 0.181]. However, at 10 years of age, the GFR values were significantly lower in children with VURD syndrome [Group A: 75.7 (71.2–85.9) mL/min/1.73 m^2^ vs. Group B: 95.1 (81.2–114.2) mL/min/1.73 m^2^, *p*-value: 0.009].

Similar results were achieved with the longitudinal analysis (Group A, *n* = 6; Group B, *n* = 33, Table 2). The 2 populations of children were, again, comparable with regard to the rate of antenatal diagnosis [Group A: *n* = 4 (66.7%) vs. Group B: *n* = 14 (42.4%), *p*-value: 0.387], age at valve ablation [Group A: 45 (27–68) days vs. Group B: 48 (15–179) days, *p*-value: 0.861], and nadir serum creatinine levels [Group A: 26.0 (23.0–31.0) µmol/L vs. 27.0 (23.0–32.0) µmol/L, *p*-value: 0.907].

As shown in Figure 3, on univariate analysis, no statistically significant difference was identified between the 2 groups of longitudinally observed patients for GFR values measured at 5 years of age [Group A: 93.1 (76.9–103.5) mL/min/1.73 m^2^ vs. Group B: 97.5 (80–113) mL/min/1.73 m^2^, *p*-value: 0.460]. The GFR values at 10 years of age were significantly lower in children with VURD syndrome [Group A: 71.9 (71.9–85.9) mL/min/1.73 m^2^ vs. Group B: 94.8 (81.5–110.6) mL/min/1.73 m^2^, *p*-value: 0.024].

## 4. Discussion

Between 15% to 20% of children with PUV present with a gross unilateral VUR associated with an ipsilateral, dysplastic, and non-functioning kidney, a condition also known as VURD syndrome [7].

It has been suggested that it occurs either because of an abnormality of development of the ureteric bud, or secondary to hydrostatic pressure from the bladder being transmitted to the developing kidney [7].

An early report published in 1982 by Hoover and Duckett first theorized that the VURD syndrome could act as a protective mechanism for the contralateral kidney, mitigating potential high-pressure effects within the bladder caused by the urethral obstruction from being transmitted to the upper tract [7]. Since this first description, several studies have investigated the effects of VURD syndrome on long-term kidney outcomes in children with PUV, however, with contradictory results.

In fact, 10 (83.3%) of the 12 PUV children with VURD syndrome included in their report presented normal serum creatinine levels, while the remaining 2 boys had only marginally increased serum creatinine levels (in the range of 1.5 mg/dL). In this initial study, however, the authors did not mention the age at which creatinine was measured as well as the duration of the follow-up, and no control groups were included in the analysis.

Among the authors who supported the protective role of this pop-off mechanism against chronic renal disease, the following year, Greenfield et al. [8] described 8 children with PUV and unilateral VUR. At a median follow-up period of 4 years, blood urea and plasma creatinine proved to be normal in 6 (75.0%) of the 8 patients included, supporting the hypothesis that the high pressures generated by the urethral obstruction were successfully dissipated into the invariably dilated and non-functioning refluxing kidney unit.

Similarly, in 1988, Rittenberg et al. [4] investigated the role of the protecting pop-off mechanisms known to result in kidney function preservation in children with PUV. Of the 9 patients who fulfilled the criteria of VURD syndrome, 8 (88.9%) showed serum creatine levels of 1 mg/dL or less while only one 13-year-old patient developed chronic kidney failure. 

Analog conclusions were reached more recently by Massaguer et al. [9] who retrospective reviewed 70 patients with PUV (14 patients of those having pop-off mechanisms) with a median follow-up of 7.4 years.

Since these initial reports, several other studies have investigated the prognostic value of the VURD syndrome, however, with controversial results.

To date, however, other authors have challenged the protecting role of VURD syndrome in long-term kidney function preservation.

In 1997, Cuckow et al. [10] performed a retrospective study including 12 patients undergoing unilateral nephrectomy for ipsilateral non-function kidneys associated with VUR. Serial serum creatinine levels and GFR values were analyzed and compared to age-matched controls. Interestingly, the estimated GFR was within the normal range in only 25% of boys tested in year 2 of life, and between ages 5 and 8 years. 

More recently, Narasimhan et al. [11] identified 13 PUV patients affected by VURD syndrome, 7 (53.8%) of whom showed scars in the contralateral kidney and suffered from recurrent urinary tract infections and diurnal incontinence. This study further supported the exclusion of VURD syndrome as a favorable kidney prognostic factor.

In line with these reports is a retrospective study published by Hogan et al. [12] run in 2014. They compared the number of children with CKD stage 3 or greater (GFR < 60 mL/min/1.73 m^2^ based on radionuclide kidney scans or serum creatinine levels) in PUV patients with (7/23, 30.4%) and without (26/66, 39.4%) VURD syndrome. On univariate analysis, no statistically significant relationship was identified between VURD syndrome and kidney impairment (HR:1.05, 95% CI: 0.65–1.70). The authors concluded that the originally hypothesized protective effect of VURD syndrome was not as significant as formerly postulated.

Similarly, a recently published analysis from Delefortie et al. [13] compared kidney function among a cohort of 137 boys with (*n* = 39) and without (*n* = 98) pop-off syndrome (including, therefore, both patients with VURD and with urinoma or urinary ascites) with a follow-up of, at least 5 years and found no statistical difference.

In stark contrast is the conclusion of Massaguer et al. [9] whose retrospective review of 70 patients (14 patients of those having pop-off mechanisms) with a median follow-up of 7.4 years, suggested that the pop-off mechanisms can be considered as a protective factor against chronic renal disease.

To go further, the results of our study not only challenge the protective role of the VURD syndrome on long-term kidney function preservation but also suggest that children with VURD syndrome can have worse GFR values at 10 years of age as a result of a higher drop in function during childhood. This was proved in the entire population analyzed (cross-section analysis) as well as in the selected cohort of patients that we followed up longitudinally. In fact, in children with unilateral renal damage, compensatory hypertrophy is generally seen, and increased filtration by the remaining functioning kidney to normalize GFR. However, this compensatory mechanism could contribute to kidney function decline.

To date, several animal and clinical studies have shown that a reduced functional nephron number results in compensatory glomerular hyperfiltration exposing children born with congenital solitary kidneys to kidney function deterioration with aging [14,15,16]. Our study seemed to confirm these results in children born with PUV and a solitary functioning kidney associated with an ipsilateral gross VUR.

Another important element to consider in the longitudinal analysis would be the presence of high bladder pressures which could play an important role in the development of VURD (as well as the other pop-off mechanisms). In their publication, D’Oro et al. [17] suggested that the development of a pop-off could indicate higher pressure built up prior to valve ablation.

With respect to previous reports on VURD syndrome, the main strength of our study is the large sample size of children followed up with a robust protocol, the strict inclusion criteria (we excluded late diagnosed patients who, we believe, represent a different population; furthermore, in order to have a thorough and robust selection criteria for identifying non-functioning/poorly functioning renal units from birth we have included only children with a nuclear medicine scan performed within the first few months of life and distant from episodes of acute urinary tract infection), the longest follow-up, and the use of the most accurate way to measure the kidney function (iohexol GFR). Measured GFR is one of the most accurate tests to certify GFR levels; an alternative test to measure renal function is Cystatin C but this is influenced by other non-GFR-related factors such as obesity, thyroid function, and cardiovascular risk factors [18].

Our longitudinal analysis provides further strength to our results.

With the limits inherent to any retrospective study, our findings suggested that not only VURD syndrome does not show a protective role in kidney function preservation, but it seems to act as a negative predictive factor.

Further studies with larger and multicenter cohorts and a longer follow-up are warranted to confirm our results and investigate whether children born with PUV and a solitary functioning kidney are at increased risk of kidney function deterioration, particularly in adolescence and adulthood.

## 5. Conclusions

Data from our study suggested that VURD syndrome does not represent a protective condition for the long-term kidney function of boys with born with PUV. On the contrary, it seemed to lead to kidney function deterioration on a long-term follow-up.

## Figures and Tables

**Figure 1 ijerph-20-06238-f001:**
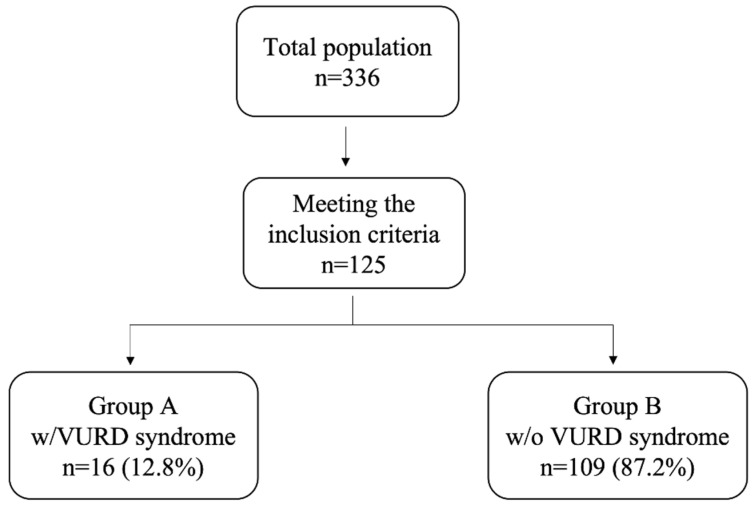
Flow chart of patients meeting the inclusion criteria and showing VURD syndrome present in almost 13% of our population.

**Figure 2 ijerph-20-06238-f002:**
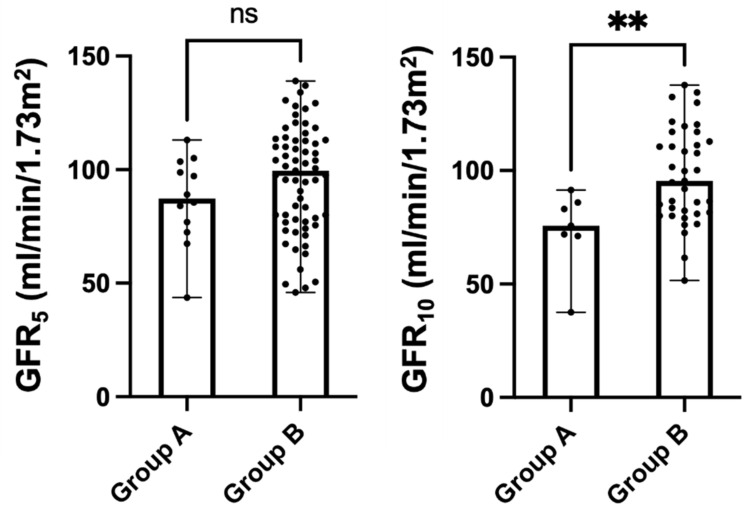
Kidney function as measured via iohexol clearance at 5 (GFR_5_) and 10 (GFR_10_) years of age in children with (Group A, *n* = 16, 12.8%) and without (Group B, *n* = 109, 87.2%) VURD syndrome. The entire cohort of patients with PUV included in the study was considered in the analysis. Scatter dot plots were represented as median with ranges. Abbr. ns: not statistically significant; ** *p*-value < 0.001.

**Figure 3 ijerph-20-06238-f003:**
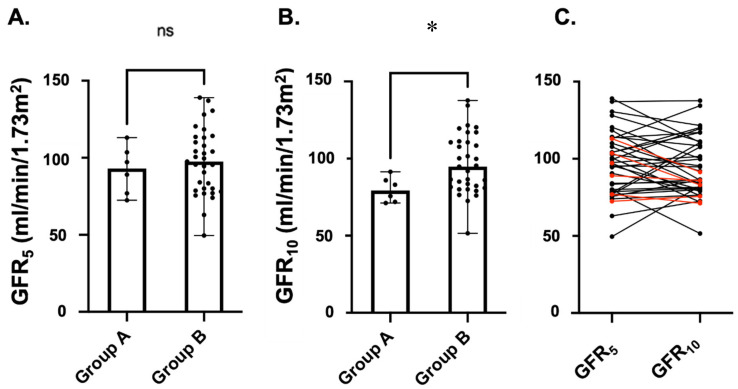
Kidney function as measured via iohexol clearance in children with (Group A, *n* = 6, 15.4%) and without (Group B, *n* = 33, 84.6%) VURD syndrome. Only children with PUV who underwent this measurement at both 5 (GFR_5_—**A**) and 10 (GFR_10_—**B**) years of age were considered in this longitudinal analysis. Scatter dot plots were represented as medians with ranges. In (**C**), we show the longitudinal change in GFR value for each patient. Red lines represent children of Group A and black lines children of Group B. Abbr. ns: not statistically significant; * *p*-value < 0.05.

**Table 1 ijerph-20-06238-t001:** Baseline characteristics of the overall patient cohort.

	Group A (*n* = 16; 12.8%)	Group B (*n* = 109; 87.2%)	*p*-Value
Antenatal diagnosis	8 (50.0%)	65 (59.6%)	0.589
Age of valve ablation (days) (median; q_1_–q_3_)	48.0 (30.5–230.5)	33.0 (16.0–108.0)	0.093
Nadir serum creatinine levels (µmol/L) (median; q_1_–q_3_)	26.5 (22.5–34.0)	26.0 (22.0–33.0)	0.935
Age at nadir serum creatinine levels (days) (median; q_1_–q_3_)	256.0 (153.5–362.0)	198.0 (125.0–369.0)	0.545
Peak serum creatinine levels (µmol/L) (median; q_1_–q_3_)	72.0 (35.5–108.0)	107.0 (70.0–215.0)	0.018
Age at peak serum creatinine levels (days) (median; q_1_–q_3_)	12.5 (3.0–187.0)	7.0 (3.0–30.0)	0.441

**Table 2 ijerph-20-06238-t002:** Baseline characteristics of the selected cohort of patients who had the GFR measured at both 5 and 10 years of age.

	Group A (*n* = 6; 15.4%)	Group B (*n* = 33; 84.6%)	*p*-Value
Antenatal diagnosis	4 (66.7%)	14 (42.4%)	0.387
Age of valve ablation (days) (median; q_1_–q_3_)	45.0 (27.0–68.0)	48 (15.0–179.0)	0.861
Nadir serum creatinine levels (µmol/L) (median; q_1_–q_3_)	26.0 (23.0–31.0)	27.0 (23.0–32.0)	0.907
Age at nadir serum creatinine levels (days) (median; q_1_–q_3_)	153.5 (145.0–166.0)	216.0 (104.0–373.0)	0.448
Peak serum creatinine levels (µmol/L) (median; q_1_–q_3_)	82.5 (41.5–137.0)	129.0 (90.0–204.0)	0.242
Age at peak serum creatinine levels (days) (median; q_1_–q_3_)	5.0 (3.0–33.0)	10.0 (3.0–30.0)	0.824

## Data Availability

Not applicable.

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
