# Peer review of "Posterior Urethral Valves, Unilateral Vesicoureteral Reflux, and Renal Dysplasia (VURD) Syndrome: Long-Term Longitudinal Evaluation of the Kidney Function"

_ijerph, 2023, doi:10.3390/ijerph20136238_

Round 1
Reviewer 1 Report
This report is an interesting retrospective study, investigating the role of VURD syndrome on long-term kidney function in children with VUR. The topic is interesting and published data are still scares. Methods and results are correctly exposed. The study is generally well conducted even if with some limitations, as also highlighted by authors.
These are my suggestions:
- I suggest an English revision of the text
- I suggest adding in the introduction, some functional details regarding the possible role of VURD, in order to facilitate the read of the paper.
- The first part of the discussion (until line 193) is a detailed review of published reports regarding the role of VURD. Even if interesting, this part is too long and hard to read. I suggest summarizing this information and put together studies that shows similar results. I also suggest discussing more in details in the discussion the potential mechanisms behnd the protective role of VURD on kidney function (see as example PMID: 24679861
- A confounding factor potentially affecting the functional outcomes of patients is the infectious risk from resistant strains affecting uropathic patients, with a possible impact on kidney function. I suggest adding information about urinary tract infections/pyelonephritis in the results, if available, and/or referring to this aspect in the discussion (see as example doi.org/10.3390/children8060436 or doi.org/10.3390/children8070597)
Author Response
This report is an interesting retrospective study, investigating the role of VURD syndrome on long-term kidney function in children with VUR. The topic is interesting and published data are still scares. Methods and results are correctly exposed. The study is generally well conducted even if with some limitations, as also highlighted by the authors.
These are my suggestions:
- I suggest an English revision of the text
One of the authors, who is a native English speaker, has re-read the revised manuscript and did not find obvious defects in the grammar or idiom.
- I suggest adding in the introduction, some functional details regarding the possible role of VURD, in order to facilitate the read of the paper.
We have added a sentence in the introduction as suggested.
“The Potential effect of VURD is to protect the function of a single renal unit by dissipating the pressure created in the obstructed bladder toward the contralateral (including the ureter) that will be acting as extra-reservoir giving up the function of the nephrons”.
- The first part of the discussion (until line 193) is a detailed review of published reports regarding the role of VURD. Even if interesting, this part is too long and hard to read. I suggest summarizing this information and put together studies that shows similar results. I also suggest discussing more in details in the discussion the potential mechanisms behind the protective role of VURD on kidney function (see as example PMID: 24679861)
We have edited the discussion, as requested.
- A confounding factor potentially affecting the functional outcomes of patients is the infectious risk from resistant strains affecting uropathic patients, with a possible impact on kidney function. I suggest adding information about urinary tract infections/pyelonephritis in the results, if available, and/or referring to this aspect in the discussion (see as example doi.org/10.3390/children8060436 or doi.org/10.3390/children8070597)
We agree the urinary infections, particularly sustained by resistant pathogens could be a confounding factor. These data were not available for our patients, and we could not add this information in the results. We have included a sentence in the discussion: “Presence of urinary infections, particularly sustained by resistant pathogens could also represent a confounding factor”.
Reviewer 2 Report
In this study, the authors aimed to investigate the role of VURD syndrome in the basis of long-term kidney function with a cross-sectional and longitudinal analysis. They used only serum creatinine levels and eGFR values when evaluating this hypothesis.
However, I do have some questions and concerns outlined below about their study.
1) It should be remembered that not only dysplasia in kidneys but also long-standing bladder dysfunction attributes to kidney failure in children with PUV.
2) The authors should discuss the difference in GFR observed at the age of 5 and 10 in the group with and without VURD?
3) As exclusion criteria, what was your rationale in determining PUV ablation after two years of age?
4) To what extent do the authors think it is sufficient to indicate that VURD syndrome has no protective effect by evaluating only Cre or eGFR levels? Do they have any data on bladder pressures, lower urinary tract symptoms, etc.? As stated, not specifying the age at which the creatinine measurement was made and the follow-up period and not including any control group in the analysis constitute important shortcomings in making this interpretation.
5) According to the results, isn't serum creatine levels of 0.8 mg/dl or higher a poor prognostic marker? Moreover, is there any other parameter that can be used to monitor the effect of pop-off mechanisms as a protective factor against chronic kidney disease?
6) A table showing the results of univariate analysis was not included in the text.
7) The findings in the study does not support the statement ‘On the contrary, the reduced nephron mass seemed to lead to kidney function deterioration on a long-term follow-up.‘.
8) Lastly, the author should include the high drop-out rate as a limitation.
Author Response
In this study, the authors aimed to investigate the role of VURD syndrome in the basis of long-term kidney function with a cross-sectional and longitudinal analysis. They used only serum creatinine levels and eGFR values when evaluating this hypothesis.
We have actually measured the GFR (using iohexol) at 5 and 10 years of age. We believe measured GFR (mGFR) is a more accurate method to assess renal function and it is more consistent than using the estimate GFR (eGFR). (Levey AS, Coresh J, Tighiouart H, Greene T, Inker LA. Measured and estimated glomerular filtration rate: current status and future directions. Nat Rev Nephrol. 2020 Jan;16(1):51-64).
We are not aware of other (better) ways of assessing kidney function.
However, I do have some questions and concerns outlined below about their study.
1) It should be remembered that not only dysplasia in kidneys but also long-standing bladder dysfunction attributes to kidney failure in children with PUV.
We do agree with the reviewer that bladder dysfunction is very important as we can imagine the VUR may have developed due to high pressure in the bladder and the question is whether resection of PUV corrects this or simply set the child up for a poorly compliant bladder.
We systematically check bladder function in all our patients and are proactive in intervening if we identify any overactivity/bladder neck dysfunction.
We have acknowledged the above point in our discussion: “Another important element to consider in the longitudinal analysis would be the presence of high bladder pressures which could play an important role in the development of VURD (as well as the other pop-off mechanisms).”
2) The authors should discuss the difference in GFR observed at the age of 5 and 10 in the group with and without VURD?
We do thank the reviewer for this point. We have added the below sentence to our discussion:
“In children with unilateral renal damage, compensatory hypertrophy is generally seen and increased filtration by the remaining functioning kidney to normalize GFR. However, this compensatory mechanism contributes to kidney function decline”.
3) As exclusion criteria, what was your rationale in determining PUV ablation after two years of age?
Our rationale for not including children diagnosed above 2 years of age is that we believe they represent a different population.
Children who are not diagnosed antenatally or within their first months of life represent a different subpopulation with, usually, less severe bladder/renal dysfunction.
Furthermore, we decided to have a thorough and robust selection criteria for identifying non-functioning/poorly functioning renal units from birth (without the interference of postnatal bladder dysfunction or UTIs) by only including children in whom a nuclear medicine scan was performed within the first few months of life and distant from episodes of acute urinary tract infection.
We added the above to our methods and discussion.
4) To what extent do the authors think it is sufficient to indicate that VURD syndrome has no protective effect by evaluating only Cre or eGFR levels?
We measured renal function using Iohexol GFR and not eGFR. Iohexol is a non-ionic contrast agent used to measure GFR based on plasma clearance.
5) Do they have any data on bladder pressures, lower urinary tract symptoms, etc.? As stated, not specifying the age at which the creatinine measurement was made and the follow-up period and not including any control group in the analysis constitute important shortcomings in making this interpretation.
We thank the reviewer for this comment.
Bladder pressure and dynamics are important points. As reported above (point 1), in our unit we systematically check bladder function in all our PUV patients and are proactive in intervening if we identify any overactivity/bladder neck dysfunction.
We have acknowledged the above point in our discussion. “Another important element to consider in the longitudinal analysis would be the presence of high bladder pressures which could play an important role in the development of VURD (as well as the other pop-off mechanisms).”
We are unclear about the comment related to the age at which creatinine measurement was made as data are shown in the tables and the follow-up period is 10 years of age as described in our methods.
With regard to the “control group” we have compared children with a diagnosis of PUV with and without VURD.
6) According to the results, isn't serum creatine levels of 0.8 mg/dl or higher a poor prognostic marker?
Nadir serum creatinine has been reported by multiple authors as prognostic factor for long term renal function (Meneghesso D, Bertazza Partigiani N, Spagnol R, Brazzale AR, Morlacco A, Vidal E. Nadir creatinine as a predictor of renal outcomes in PUVs: A systematic review and meta-analysis. Front Pediatr. 2023 Mar 15;11:1085143; Delefortrie T, Ferdynus C, Paye-Jaouen A, Peycelon M, Michel JL, Dobremez E, El Ghoneimi A, Harper L. Nadir creatinine predicts long-term bladder function in boys with posterior urethral valves. J Pediatr Urol. 2022 Apr;18(2):186.e1-186.e4)
For this manuscript and for the purpose of investigating the impact of VURD syndrome we decided to measure patients’ renal function in the mid and long-term follow-up (5 and 10 years of age) utilising Iohexol GFR.
Moreover, is there any other parameter that can be used to monitor the effect of pop-off mechanisms as a protective factor against chronic kidney disease?
We believe Iohexol GFR is a robust and validated method for monitoring the changes in renal function. (Levey AS, Coresh J, Tighiouart H, Greene T, Inker LA. Measured and estimated glomerular filtration rate: current status and future directions. Nat Rev Nephrol. 2020 Jan;16(1):51-64)
7) A table showing the results of univariate analysis was not included in the text.
The results of the univariate analysis are described in the “results” section and in Table 1.
8) The findings in the study does not support the statement ‘On the contrary, the reduced nephron mass seemed to lead to kidney function deterioration on a long-term follow-up.‘
We have removed the statement in our conclusions.
9) Lastly, the author should include the high drop-out rate as a limitation.
We explained the rationale of not including children diagnosed above 2 years of age as we believe they represent a different population; furthermore, in order to have a thorough and robust selection criteria for identifying non-functioning/poorly functioning renal units from birth we have included only children with a nuclear medicine scan performed within the first few months of life and distant from episodes of acute urinary tract infection. This has been added in the “methods” section.
Round 2
Reviewer 2 Report
- Cystatin C is used as a better way of kidney function throughout different institutions, it is at least to mention in the limitations. Please take a look at https://www.nejm.org/doi/full/10.1056/nejmoa1214234
Author Response
We thank the reviewer for his comment but we would like to point out that we have utilized Iohexol plasma clearance which is one of the best tests to measure renal function. To reflect the above, we have added a sentence to our discussion: "Measured GFR is one of the most accurate tests to certify GFR levels; an alternative test to measure renal function is Cystatin C but this is influenced by other non-GFR-related factors such as obesity, thyroid function, and cardiovascular risk factors. {Delanaye P, Melsom T, Ebert N, Bäck SE, Mariat C, Cavalier E, et al. Iohexol plasma clearance for measuring glomerular filtration rate in clinical practice and research: a review. Part 2: why to measure glomerular filtration rate with iohexol? Clin Kidney J. 2016;9(5):700–4.}".